# Predictable and stable epimutations induced during clonal plant propagation with embryonic transcription factor

Anjar Tri Wibowo[1,2,3☯], Javier Antunez-Sanchez[1☯], Alexander Dawson[1], Jonathan Price[1], Cathal Meehan[1], Travis Wrightsman[2], Maximillian Collenberg[2], Ilja Bezrukov[2], Claude Becker[2,4], Moussa Benhamed[5], Detlef Weigel[2]*, Jose Gutierrez-Marcos[1]*

1 School of Life Science, University of Warwick, Coventry, United Kingdom, 2 Department of Molecular Biology, Max Planck Institute for Developmental Biology, Tubingen, Germany, 3 Department of Biology, Faculty of Science and Technology, Airlangga University, Surabaya City, East Java, Indonesia, 4 Ludwig-Maximilians-University of Munich, Faculty of Biology, Biocenter, Martinsried, Germany, 5 Université Paris-Saclay, Centre National de la Recherche Scientifique, Institut National De La Recherche Agronomique, University of Évry, Institute of Plant Sciences Paris-Saclay (IPS2), Orsay, France

☯ These authors contributed equally to this work.
* detlef.weigel@tuebingen.mpg.de (DW); j.f.gutierrez-marcos@warwick.ac.uk (JGM)

**Data Availability Statement:** Sequence data (BS-seq, RNA-seq and DNA-seq) and Ws-2 genome assembly have been deposited at the European

## Abstract

Clonal propagation is frequently used in commercial plant breeding and biotechnology programs because it minimizes genetic variation, yet it is not uncommon to observe clonal plants with stable phenotypic changes, a phenomenon known as somaclonal variation. Several studies have linked epigenetic modifications induced during regeneration with this newly acquired phenotypic variation. However, the factors that determine the extent of somaclonal variation and the molecular changes underpinning this process remain poorly understood. To address this gap in our knowledge, we compared clonally propagated *Arabidopsis thaliana* plants derived from somatic embryogenesis using two different embryonic transcription factors- *RWP-RK DOMAIN-CONTAINING 4 (RKD4)* or *LEAFY COTYLEDON2 (LEC2)* and from two epigenetically distinct founder tissues. We found that both the epi (genetic) status of the explant and the regeneration protocol employed play critical roles in shaping the molecular and phenotypic landscape of clonal plants. Phenotypic variation in regenerated plants can be largely explained by the inheritance of tissue-specific DNA methylation imprints, which are associated with specific transcriptional and metabolic changes in sexual progeny of clonal plants. For instance, regenerants were particularly affected by the inheritance of root-specific epigenetic imprints, which were associated with an increased accumulation of salicylic acid in leaves and accelerated plant senescence. Collectively, our data reveal specific pathways underpinning the phenotypic and molecular variation that arise and accumulate in clonal plant populations.

Nucleotide Archive (ENA) under BioProject
PRJNA819891

**Funding:** This work was supported by ERA-CAPS
AUREATE (DFG) and the Max Planck Society to
(DW), an European Research Council (ERC) Marie
Skłodowska-Curie Fellowship (751204-H2020-
MSCA-IF-2016) and Airlangga University Hibah
Riset Mandat Grant (387/UN3.14/PT/2020) to
(ATW), and BBSRC grants (BB/L003023/1, BB/
N005279/1, BB/N00194X/1 and BB/P02601X/1) to
(JGM). The funders had no role in study design,
data collection and analysis, decision to publish, or
preparation of the manuscript.

**Competing interests:** The authors have declared
that no competing interests exist.

## Author summary

Clonal propagation is commonly used in plants to propagate selected genotypes and to
aid genetic/genomic manipulation. Although cloning minimizes genetic variation, clonal
plants commonly display phenotypic changes, a phenomenon known as somaclonal varia-
tion, that are at times stable after multiple cycles of sexual reproduction. The newly
acquired phenotypic variation exhibited by clonal plants has been linked to non-genetic
modifications induced during the regeneration process, though the precise nature of these
modifications and the factors implicated remain poorly understood. By generating clonal
plants through two different methods and using two distinct tissues–roots and leaves–, we
show that the phenotypic variation found in clonal plants is linked to heritable changes in
DNA methylation marks already present in the founder tissues. Further, these novel phe-
notypes are associated with specific patterns of gene expression and metabolic changes.
Our findings unveil specific pathways underpinning new phenotypic and molecular varia-
tion that arises and accumulates in clonal plant populations.

## Introduction

Many multicellular organisms can reproduce both sexually and asexually, with clonal repro-
duction being especially common in plants, where it is prevalent in ferns, mosses and many
angiosperms [1, 2]. Most plant species can switch to asexual reproductive programs in
response to environmental stimuli by forming specialized structures such as rhizomes, stolons
and bulbils [2]. Such flexibility in reproductive strategies is thought to be possible due to the
high degree of plasticity and totipotency of plant cells. This property has been traditionally
exploited for the clonal propagation and genomic manipulation of many economically impor-
tant species [3]. Clonal propagation can be achieved in different ways, from simple methods
such as grafting and cuttings to more advanced methods such as tissue culture-induced regen-
eration using phytohormones or embryonic/meristematic transcription factors. In *Arabidopsis
thaliana* (hereafter, Arabidopsis), somatic embryogenesis (SE) and plant regeneration can be
initiated through the ectopic overexpression of embryonic transcription factors such as *AGA-
MOUS- LIKE15 (AGL15)* [4], *BABY BOOM (BBM)* [5], *LEAFY COTYLEDON2 (LEC2)* [6],
*WUSCHEL* (*WUS*) [7–9] and *RWP-RK DOMAIN-CONTAINING 4 (RKD4)* [10].

A key characteristic of asexual propagation is that the individuals produced are genetically
identical to their parents, except for a few spontaneous *de novo* mutations [1]. However, clonal
plants often display heritable phenotypic variation [11, 12], known as somaclonal variation,
which is linked to genetic [13, 14] and epigenetic changes [12, 15, 16]. Moreover, this pheno-
typic diversity could contribute to the adaptation to recurrent environmental challenges. In a
previous study we found that Arabidopsis plants regenerated through the ectopic activation of
an embryonic transcription factor, resulted in clonal plants that inherited molecular signa-
tures, DNA methylation and gene expression profiles, characteristic of the tissue of origin [12].
Whether this molecular and phenotypic variation is primarily caused by the direct regenera-
tion of clonal individuals from different above and belowground organs, or driven by the
organ-specific activity of the embryonic transcription factor employed, has been unclear. To
address this gap in our knowledge, we generated clonal progeny from Arabidopsis roots and
leaves through the ectopic activation of two different embryonic transcription factors. We
found that the founder organ employed for cloning significantly contributes to clonal progeny,
with root regenerants inheriting tissue-specific DNA methylation imprints and transcriptional
states, independent of the regeneration method used. Results from our study reveals avenues

for the manipulation of phenotypic traits through cloning for the adoption in crop improvement programs.

## Results

### Phenotypic variation present in clonal plants is underpinned by tissue of origin

In a previous study, we identified heritable molecular and phenotypic changes in Arabidopsis Col-0 plants propagated through the induction of the embryonic transcription factor RKD4 [12]. To better understand how this somaclonal variation is created, in particular the importance of the genetic background and/or the transcription factor employed, we generated clonal progeny from Arabidopsis Ws-2 leaf and root tissues using transgenic lines that allow the controlled expression of either *RKD4* or *LEC2*, two transcription factor gene necessary for the initiation of different stages of embryo development [6, 10]. To determine the stability of the phenotypes arising during cloning, we propagated seeds from each regenerated individual (G0) by selfing over three consecutive generations (G1-G3) under similar growth conditions, thus limiting possible effects caused by tissue culture. When we grew G2 and G3 clonal progeny, we noticed that the morphology of plants regenerated from roots using LEC ectopic expression (RO-LEC2) was noticeably different to control plants, while leaf regenerants (LO-LEC2) were indistinguishable from the control plants (Fig 1A). Notably, most RO-LEC2 lines (6/10) had reduced biomass, smaller leaf area and senesced early (Fig 1C and 1D), with different temperature growth conditions having no discernible effects on these phenotypes (S1 Fig). These phenotypes were only found in RO-LEC2 lines and plants regenerated using RKD4, either from root or leaf, did not produce any observable phenotypes. Collectively, our data suggest that the tissue of origin employed for regeneration and the regeneration protocol could contribute to phenotypic diversity in the clonal plants and their progenies, thus pointing to the long-term inheritance of somatic epigenetic states in the sexual progeny of clonal plants.

### Tissue of origin affects the activity of defense-related genes

To determine what molecular changes might underpin the observed morphological variation, we performed whole-genome transcriptome analysis in leaves and roots from randomly selected, independent G3 lines. Using stringent thresholds [false discovery rate (FDR) <0.05; absolute log-two fold change >1.5], we obtained lists of Differentially Expressed Genes (DEGs) by comparing leaves and roots of regenerants and control plants (S1 and S2 Tables). There were notable differences between the regenerants induced by the two embryonic transcription factors; leaves of RKD4-induced regenerants were the most affected independent of the tissue of origin (401 and 253 DEGs in leaves for LO and RO respectively, compared to 147 and 48 in roots). Notably, we found more DEGs for LO than for RO plants (Figs 2A and S2), which resembled the effects found in Col-0 plants cloned using RKD4 [12]. LEC2 regenerants had the highest numbers of DEGs in the opposite tissue from which the plant was regenerated, i.e., in roots of LO-LEC2 plants (282 DEGs versus 75 in leaves) and in leaves of RO-LEC2 plants (595 DEGs versus 61 in roots) (Figs 2A and S3). Since gene expression was strongly dysregulated in the leaves of both RO-RKD4 and RO-LEC2 regenerants, we looked for an overlap between the DEGs. We found more shared dysregulated genes than expected by chance (p-value = $2.1 \times 10^{-17}$), suggesting common effects independent of the regeneration strategy.

To obtain further insight into the possible function of these DEGs, we performed Gene Ontology (GO) analyses, finding that DEGs in leaves from both RO-LEC2 and RO-RKD4

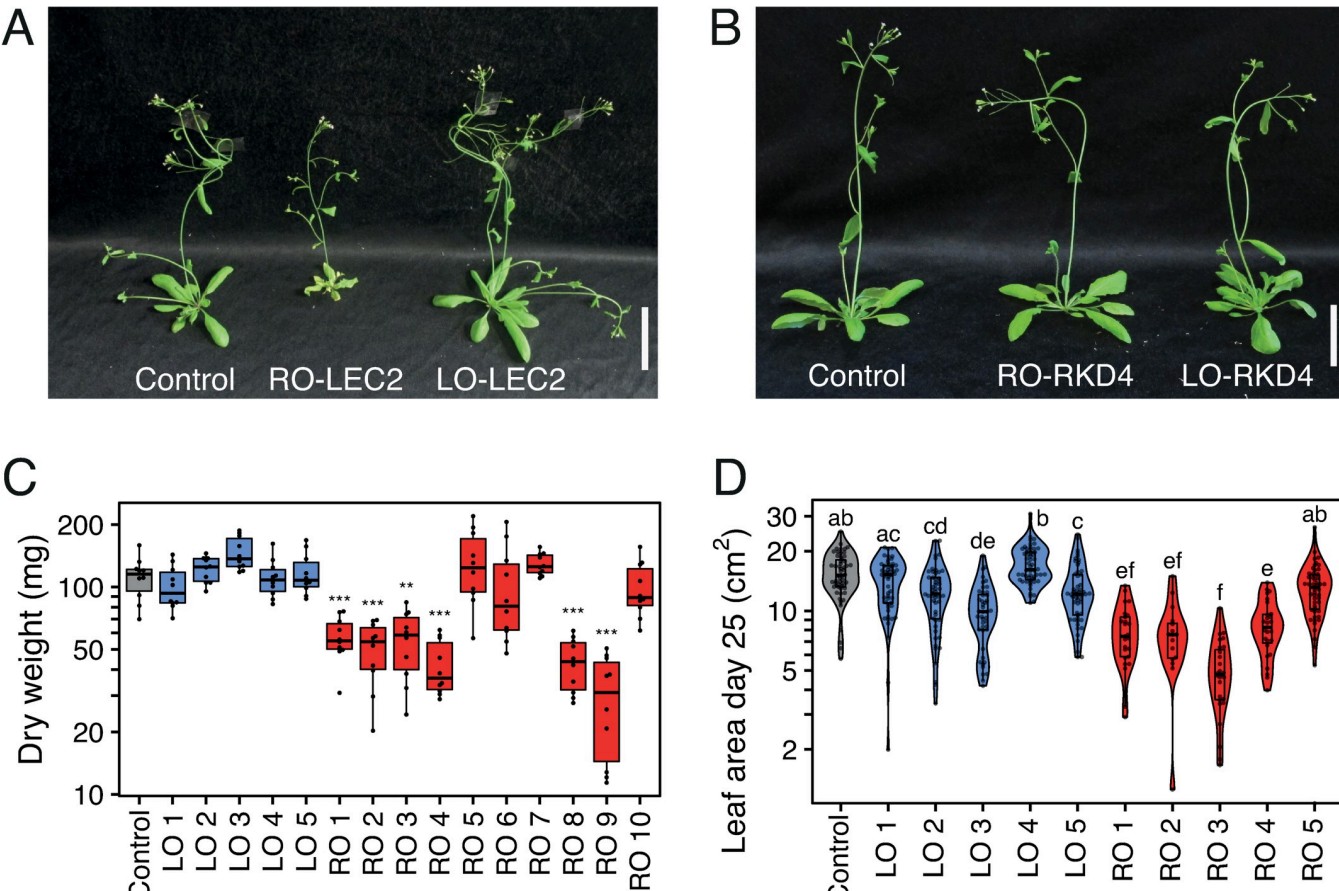

**Fig 1. Growth phenotypes of Arabidopsis plants cloned from roots (RO) and leaves (LO) through LEC2/RKD4-induced regeneration and propagated by self-fertilization over three generations.** (A) Representative images of 28-day old Arabidopsis Ws-2 plants. Control (Ws-2/indLEC2), and clonal RO-LEC2 and LO-LEC2 G3 progeny. Scale bars, 5 cm. (B) Representative images of 28-day old Arabidopsis Ws-2 plants. Control (Ws-2/indRKD4), and clonal RO-RKD4 and LO-RKD4 G3 progeny. Scale bars, 5 cm. (C) Boxplot showing variation in plant biomass for control (indLEC2/Ws-2) and independent G3 lines. Asterisks represent significant differences, determined by Student's *t-test*, ** p<0.01, *** p<0.001, sample size n = 10. (D) Violin plot and boxplot of leaf area for control (Ws-2/indLEC2) and independent G3 lines. Letters represent groups of statistically significantly different samples (p<0.05), determined by Tukey's test, sample size n = 20.

lines were enriched for stress and defense responses (FDR<0.05, Fig 2C). In contrast, DEGs in roots from either LO-LEC2 or LO-RKD4 lines were primarily associated with metabolic processes (Fig 2C). Network analysis revealed that genes related to pathogen responses and senescence were upregulated in RO plants, independent of the embryonic transcription factor used for clonal propagation (Fig 2D and 2E). As several of the genes upregulated in RO-LEC2 leaves were part of the salicylic acid (SA) biosynthesis pathway (Fig 2F), we measured the concentration of total SA in G2 progenies of LEC2 regenerated plants. Reduced biomass and early senescence in RO-LEC2 lines were strongly associated with high levels of SA, indicating a causal relationship between transcriptional changes, differences in production of a defense hormone known to negatively affect growth and to induce early senescence [17], and morphological changes in the regenerants (Fig 2G). The senescence phenotypes observed are likely caused by the disturbance of free SA and conjugated SA. Collectively, our analyses revealed that the clonal propagation of plants using embryonic transcription factors results in the transmission of organ-specific transcriptional states.

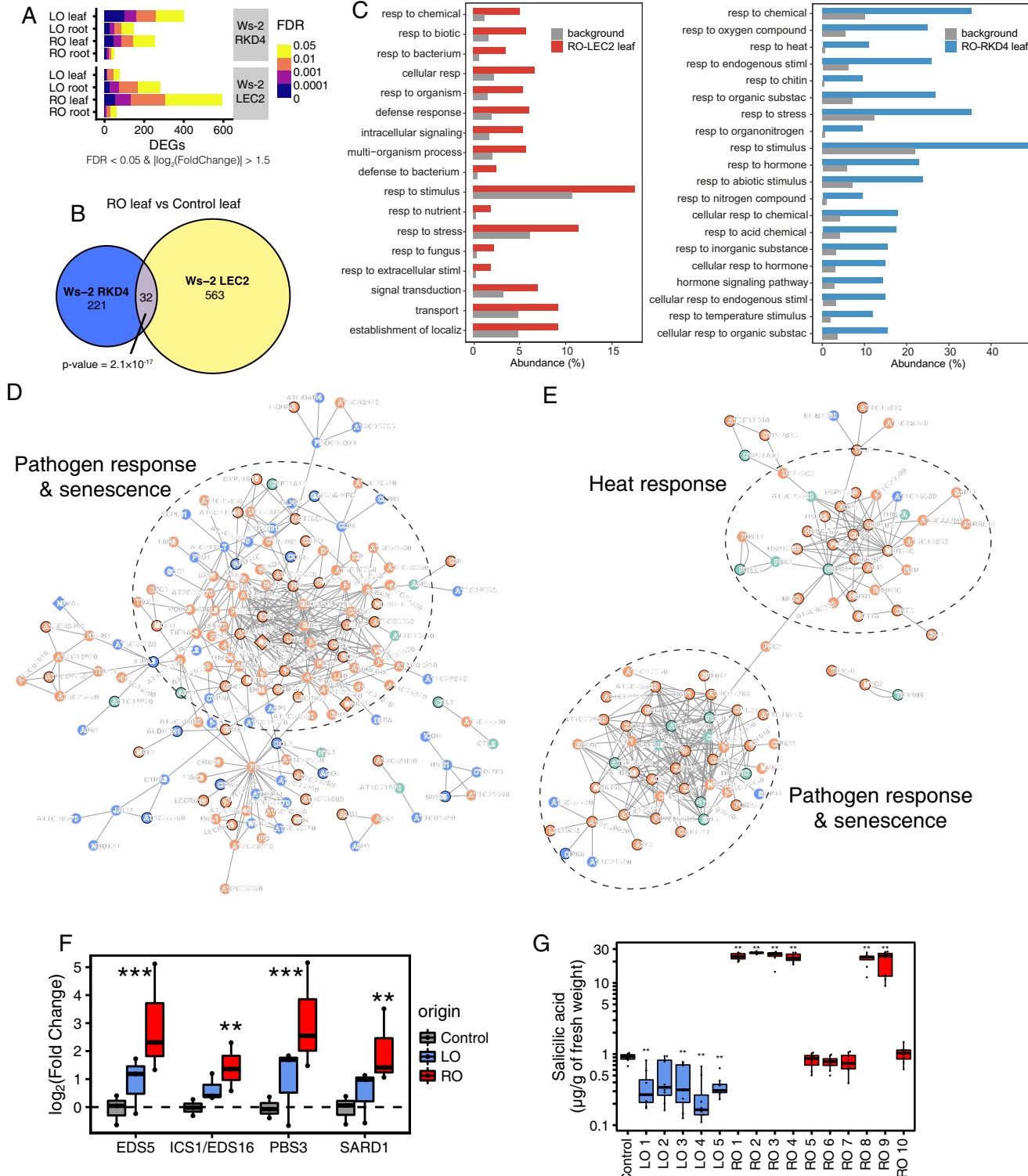

**Fig 2. Differential expression of pathogen response and senescence genes in Arabidopsis Ws-2 plants regenerated from roots or leaves using RKD4 or LEC2.** (A) Number of DEGs for different comparisons split by FDR levels. Data from three randomly selected regenerants for each tissue. (B) Euler diagram of common DEGs found in RO plants regenerated using two different transcription factors. p-value indicates results from Fisher's exact test. (C) Gene Ontology analysis showing significant enrichment for stress-related genes among DEGs in RO-LEC2 (left) and RO-RKD4 (right) leaves. Categories are sorted by decreasing FDR. (D) Interaction network of DEGs found in leaves of RO-LEC2 plants. Orange, upregulated genes; blue, downregulated genes; green, non-

DEGs with significant interactions with the network. Diamond shaped nodes, transcription factors. Main GO terms associated with clusters are highlighted. (E) Interaction network of DEGs found in leaves of RO-RKD4 plants. Orange, upregulated genes; blue, downregulated genes; green, non-DEGs with significant interactions with the network. Diamond shaped nodes, transcription factors. Main GO terms associated with clusters are highlighted. (F) Expression levels from RNA-seq of genes associated with SA biosynthesis in LO-LEC2 and RO-LEC2 plants compared to control (Ws-2/indLEC2) parent. Asterisks represent significant differences, determined by Wald test and adjusted with the Bonferroni–Holm method: ** $p < 0.01$, *** $p < 0.001$, (G) SA concentrations in leaves of LEC2 regenerants at the G3 generation. Asterisks represent significant differences, determined by Student's *t-test*, ** $p < 0.01$, sample size of n = 10. Control = Ws-2/indLEC2 parent.

## Heritable DNA methylation imprints in LEC2-induced clonal progeny

We postulated that the phenotypic and transcriptional changes found in plants regenerated with embryonic transcription factors may be caused by the inheritance of heritable tissue-specific epigenetic imprints. To test this hypothesis, we studied the methylome of LEC2 regenerants that had been propagated for two successive generations (G2 and G3). To map methylome data we generated a *de novo* assembly of the Ws-2 genome (see methods). In an all-against-all comparison of RO-LEC2 and LO-LEC2 plants, we found 226, 120 and 386 Differentially Methylation Regions (DMRs) in CG, CHG and CHH contexts, respectively (S3 Table). Principal component analysis (PCA) of the 226 CG-DMRs showed that the main source of variance in our dataset was the tissue analyzed (69% of variance), with methylome profile of leaves of RO-LEC2 plants being slightly skewed towards root samples (Fig 3A). Another 6% of variance can be explained by the tissue of origin, separating RO-LEC2 and LO-LEC2 samples from each other and from the controls, independent of the tissue sampled. Some of the remaining variance (3%) comes from the regeneration process independent of tissue sampled or founder tissue used for cloning (Fig 3A). These data suggest that even though the methylome of regenerated plants mainly corresponds to the characteristic methylome of the tissue sampled, a number of epigenetic imprints are maintained from the tissue employed for cloning, even after plants had undergone three cycles of sexual reproduction. Our data also show that regeneration by itself contributes to discrete methylation imprints that are stably inherited in clonal plants (S4 Fig). Similar, albeit weaker patterns were observed for non-CG DMRs (S4 Fig). To define the potential impact of these epigenetic changes, we mapped DMRs to genome features, which revealed that regions flanking protein coding genes were particularly likely to be associated with methylation changes (S5 Fig).

When only considering DMRs distinguishing leaves and roots of non-regenerated plants, RO-LEC2 leaves, although distinct from other samples, resembled the methylation of root samples (Fig 3B). Most DMRs in leaves of RO-LEC2 regenerants were primarily hypomethylated compared to leaves of control plants, resembling the methylation profile of these regions in control roots, and they were stably inherited (Figs 3C and S4). Similar to roots of control plants, these DMRs were also hypomethylated in roots of LO-LEC2 regenerants, thus pointing to active DNA demethylation in Arabidopsis roots. Collectively, our analyses demonstrate that induction of somatic embryogenesis and subsequent regeneration results in tissue-specific epigenetic imprints that are stable during sexual reproduction.

## Precocious leaf senescence in root-regenerants caused by epigenetic deregulation of the salicylic acid pathway

To determine the molecular basis for the early senescence phenotype observed in RO regenerants, we performed reciprocal crosses between Ws-2 and RO-LEC2 plants. Five reciprocal $F_1$ hybrids were self-pollinated to generate $F_2$ seed progeny (Fig 4A). In the $F_2$ generation, some plants were early senescing, characteristic of RO-LEC2 plants, while others were not. The fraction of early-senescing plants was around one quarter, with 16% (13 in 80) early-senescing $F_2$

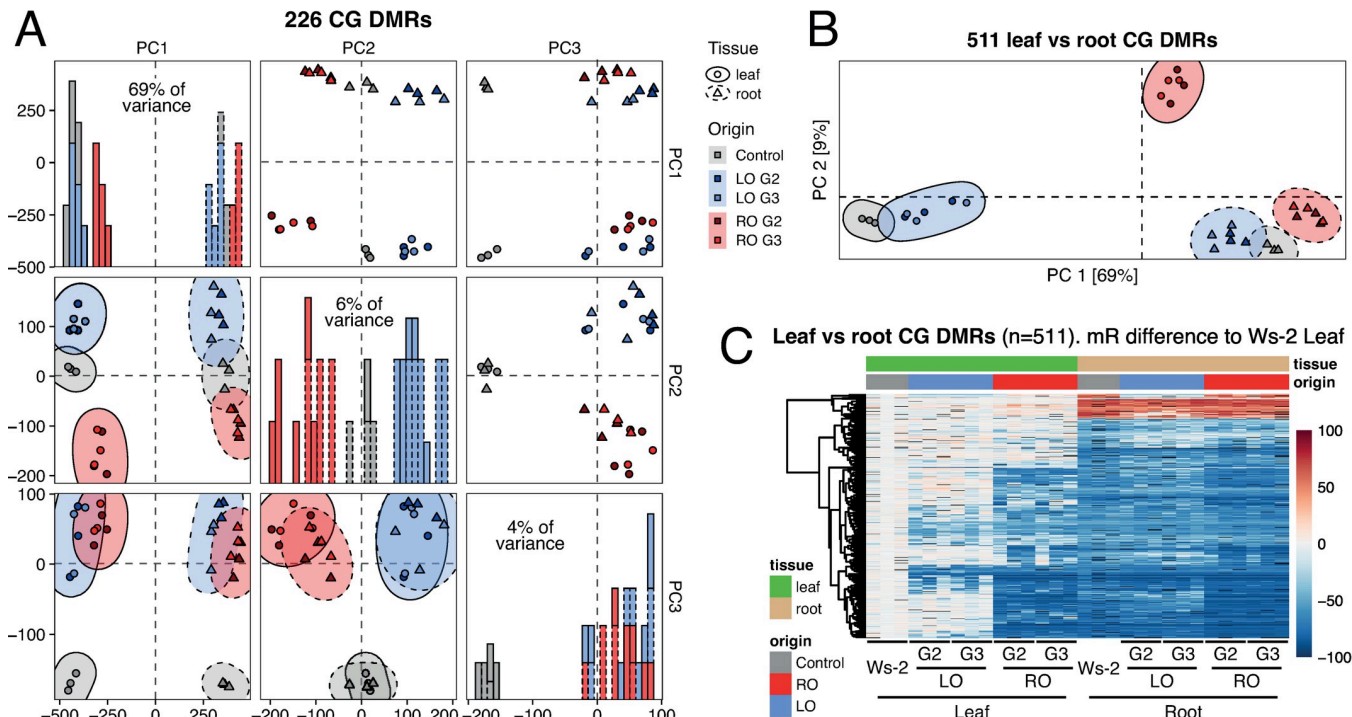

**Fig 3. DNA methylomes of LEC2 regenerants.** (A) PCAs of CG DNA methylation levels at 226 CG-DMRs identified in all-against-all comparison and histograms of the distribution of samples across each principal component. Probability ellipses (95% distribution) for each group are shown. Gray, control; Blue, LO-LEC2; Red, RO-LEC2. Data from three randomly selected regenerants for each tissue. (B) PCA of CG methylation levels at 511 CG-DMRs identified in the comparison of control leaves and roots. Probability ellipses (95% distribution) for each group are shown. Gray, control; Blue, LO-LEC2; Red, RO-LEC2. Data from three randomly selected regenerants for each tissue. (C) Heatmap of CG methylation levels in leaves and roots of control plants and LEC2 regenerants. Each column corresponds to an independently regenerated line. Ws-2 = Ws-2/indLEC2. Data from three randomly selected regenerants for each tissue.

progeny derived from Ws-2/indLEC2 x RO-LEC2 hybrids, and 23% (28 in 120) early-senescing $F_2$ progeny derived from RO-LEC2 x Ws-2/indLEC2 hybrids (p>0.05, Fisher's Exact test). These data suggest that early senescence in RO-LEC2 regenerants may be under the control of a single recessively acting locus. To determine whether this was due to a genetic mutation, we whole-genome shotgun sequenced 10 phenotypically normal and 10 early-senescing individuals from five different segregating $F_2$ populations (Fig 4A). However, we could not identify genetic variants linked to this phenotype (S6 Fig), suggesting that the phenotypes were caused by an epigenetic change.

We investigated the mode of inheritance of RO imprints by generating DNA methylation profiles of leaves from $F_2$ progeny derived from Ws-2/indLEC2 x RO-LEC2 hybrids. If differences in DNA methylation were responsible for the growth phenotypes characteristic of RO plants, we reasoned the methylation imprints must be already present in the parents. We therefore focused on DMRs between leaves of Ws-2/indLEC2 and RO-LEC2 plants. Most regions that are hypomethylated in RO-LEC2 regenerants were stably inherited both mitotically and meiotically (Fig 4B). On the other hand, most hypermethylated DMRs had disappeared after two generations of paternal transmission, although they were faithfully inherited maternally (Fig 4B). This was observed for both CG- and CHG-hypermethylated DMRs, while CHH-hypermethylation was reset both paternally and maternally (S7 Fig). These data suggest that the early senescence phenotype observed in RO-LEC2 plants may be caused by an epimutation associated with a reduction in DNA methylation, as plants with early senescence

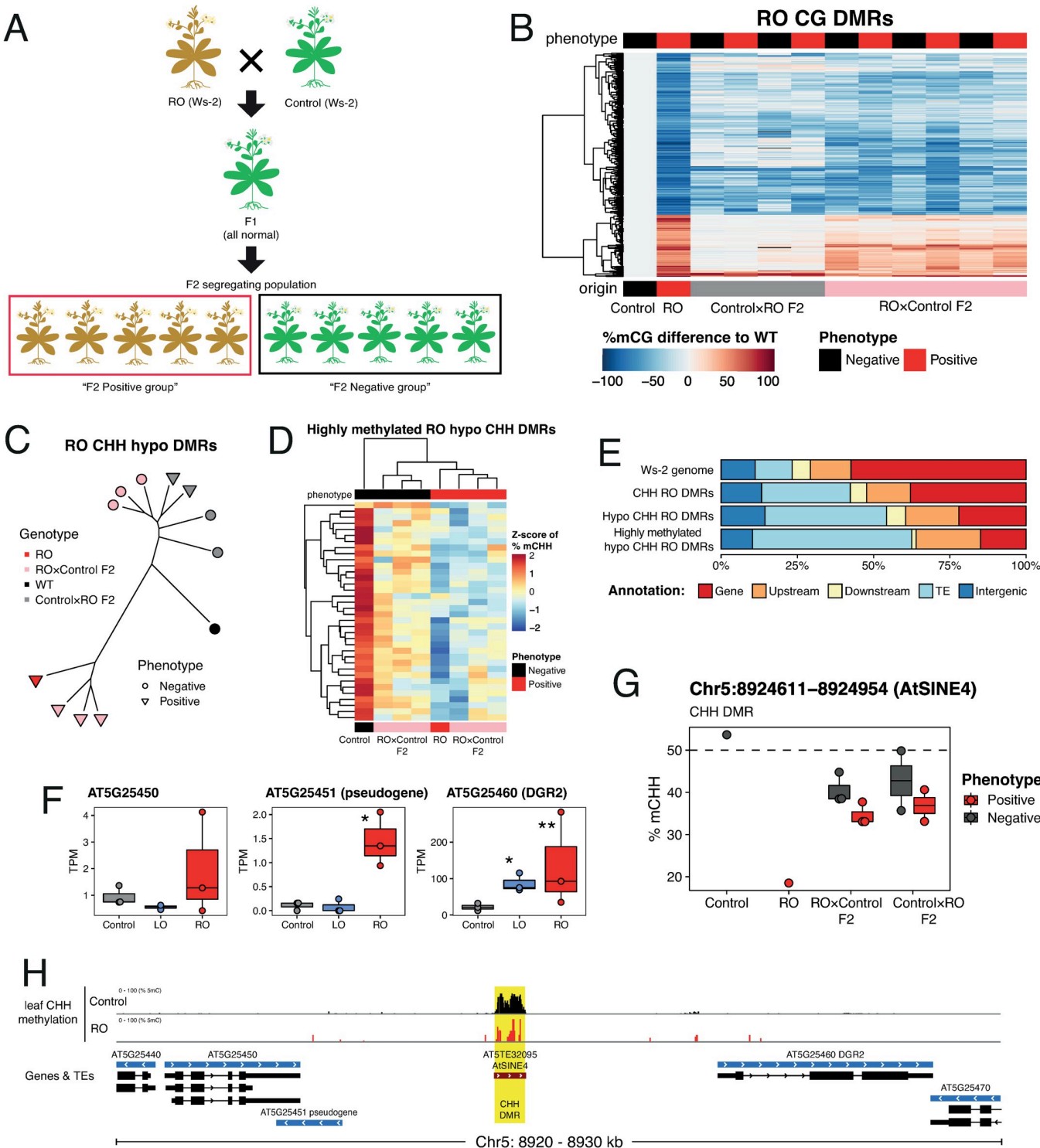

**Fig 4. Leaf senescence and TE hypomethylation in RO-LEC2 root regenerants.** (A) Experimental design to identify epi- and DNA mutations linked to RO-LEC2 phenotypes. Green indicates phenotypically normal plants (n = 10) and brown early senescing ones (n = 10). Analysis was carried out using five independent F$_2$ progeny. (B) Heatmap showing gains and losses of CG methylation for RO-LEC2-leaf DMRs in F$_2$ progeny from reciprocal backcrosses between control (Ws-2/indLEC2) and RO-LEC2 plants. Each column represents an independent F$_2$ progeny individual. (C) Dendrogram of F$_2$ progeny and parents at parental CHH-hypo-DMRs. (D) Heatmap of Z-scores in CHH methylation for highly methylated (Ws-2/indLEC2%mC > 50) parental CHH-DMRs, showing clustering of parents and RO x Ws-2/indLEC2 F$_2$ progeny. (E) Annotation of several subsets of parental CHH DMRs to different genetic features compared to the distribution of features in the whole genome. Upstream and downstream are defined as 2 kb away from a gene and Intergenic as all sequences

more than 2 kb away from any gene. The priority of annotations is Gene, TE, Upstream, Downstream and Intergenic in that order. (F) Expression levels from RNA-seq for genes proximal to Chr5:892411–894943 (AtSINE4) DMR in leaves of control (Ws-2/indLEC2) and regenerants. Asterisks represent significant differences, determined by Wald test and adjusted with the Bonferroni–Holm method: * p < 0.05, ** p < 0.01 (G) CHH methylation levels for Chr5:892411–894943 (AtSINE4) DMR in Ws-2/indLEC2, RO-LEC2, and $F_2$ reciprocal crosses, showing association between demethylation and morphological phenotype. (H) Genomic view around Chr5:892411–894943 (AtSINE4) DMR. Yellow box represents RO-LEC2 CHH-DMR.

appeared in both directions of the cross. After clustering samples based on the levels of methylation for DMRs, we found that CHH-hypomethylated DMRs as a class predicted phenotype, (Fig 4C), and this was clearest for RO-LEC2 x Ws-2/indLEC2 progeny. Thirty-six DMRs co-segregated with the growth phenotype. Normal plants were highly methylated, as in the Ws-2/indLEC2 control, and early senescing plants had low methylation at these 36 DMRs (Fig 4D). These phenotype-associated DMRs were strongly enriched in transposable elements (TEs) (Fig 4E). Collectively, these data suggest that demethylation of TEs is associated with early senescence of RO-LEC2 plants.

To understand how losses of DNA methylation at specific loci could cause the phenotype, we searched for deregulated genes in the proximity of the 36 DMRs. We found two genomic regions where methylation status co-segregated with the phenotype for both directions of the crosses (Fig 4G). One region included three genes in its proximity that are upregulated in leaves of RO-LEC2 regenerants (Fig 4F). The DMR is located in a TE of the AtSINE4 family, between two of the genes found to be misregulated in RO-LEC2 plants when compared to Ws-2/indLEC2 control (Fig 4H). We investigated the chromatin conformation of these regions in leaves and roots of wild-type plants, and found notable differences, thus suggesting it may act as a long-distance regulatory element for flanking genes (S8 Fig). Another DMR associated with early senescence in RO-LEC2 plants was located in an AtMUNX1 TE upstream of the *PRE1/BNQ1* locus (S9 Fig), in a region flanking a CarG box known to be a target of MADS box containing transcription factors [18] and negatively regulated by the transcription factor pair APETALA3 (AP3) and PISTILLATA (PI) [19].

## Discussion

A common concern surrounding clonal propagation in plants by tissue culture is the appearance of new undesirable phenotypes. This outcome may be in part attributed to the type of regeneration employed ie. organogenesis or somatic embryogenesis. For instance, using phytohormones is known to cause stochastic changes in DNA methylation, primarily hypomethylation, that are associated with changes in gene expression [11, 13, 16, 20, 21]. On the other hand, regeneration via somatic embryogenesis using embryonic transcription factors results in nonrandom changes in DNA methylation, which are heritable and associated with transcriptional and phenotypic variation [12]. To define the factor(s) that contribute to the molecular and phenotypic diversity found in plants propagated through somatic embryogenesis, we conducted a detailed analysis of regenerants created either by the ectopic expression of LEC2 or RKD4, two transcription factors that are part of distinct embryonic transcriptional networks [10, 20, 22]. Through this approach, we were able to identify heritable tissue-specific DNA methylation imprints in these regenerants that were associated with specific patterns of gene expression, leading to significant phenotypic variation. Notably, the molecular and phenotypic variation observed in clonal plants was independent of the transcription factor used for regeneration and instead strongly linked to the founder tissue employed.

The molecular mechanism(s) underpinning the inheritance of tissue-specific DNA methylation imprints that accumulate in these clonal plants are yet unknown, but they may be linked to the capacity of these two transcription factors to reprogram somatic cells such that they

adopt an embryonic program without altering their epigenomic configuration. One explanation for this phenomenon could be that tissue-specific imprints arising during cloning are developmentally programmed, since leaves and roots have distinct epigenetic characteristics [23–27]. The tissue-specific imprints found in clonal plants may be linked to the distinct chromatin architecture found in root and shoot cells, which is largely influenced by chromatin loops associated with repressive histone marks [28]. Chromatin interactions likely contribute to the developmental differentiation of specific cell types in plants, as is the case in animals [29, 30], and it would be interesting to test if these interactions are maintained or reset in clonal plants. Nevertheless, our data show that some epigenetic imprints can be fixed independently of the pathway recruited to achieve somatic embryogenesis and regeneration. This suggests that genetic and epigenetic programs can be uncoupled in clonal plants.

Our data further show that irrespective of the transcription factor employed, plants regenerated from roots senesced early–a phenotype that was inherited by the progeny either after selfing or outcrossing. The observed early leaf senescence strongly correlated with DNA hypomethylation of specific TEs flanking genes associated with the SA pathway. Our data suggest that epimutations at TEs could accumulate in populations of species that reproduce asexually, and that these epimutations could confer an adaptive advantage under fluctuating environments. Several studies have shown that the long-distance regulatory action of TEs in Arabidopsis is associated with changes in DNA methylation [31]. Moreover, transposon mobilization and variation in TE methylation have been linked to environmental adaptation in natural plant populations [32] and that this epigenetic variation often occurs near genes associated with environmental responses [33, 34]. As such, the methylation of transposons has been selected to provide new regulatory functions in most eukaryotic genomes [35].

The SA pathway may be an important target for the accumulation of epimutations in plants, as it is critical for defense [36], and is strongly correlated with biomass, a characteristic that has traditionally been selected by breeders to increase yield in various crops, including maize [37], rice [38], and wheat [39]. Notably, mutants that cause a global change in chromatin organization, such as *CROWDED NUCLEI* (*CRWN*), display constitutive activity of the salicylic acid-dependent pathway of immunity, with *crwn* plants displaying dwarfism and premature senescence [40, 41].

Epigenetic imprints generated during cloning can be inherited over multiple cycles of sexual reproduction, suggesting that the epigenetic machinery in plants is unable to repair these epimutations [11, 12, 16]. The stable accumulation of epimutations has also been documented in crosses between wild-type plants and mutants defective in DNA methylation [42–46] and histone demethylation [47]. Notably, while the tissue-specific DNA hypomethylation imprints of clonal plants are transmitted through both the maternal and paternal germline to the offspring. However, DNA hypermethylation imprints are only transmitted maternally. The molecular mechanisms implicated in the resetting of these DNA hypermethylated imprints are not yet known but may be linked to the active epigenomic reprogramming that takes place during male gametogenesis [48, 49] and/or embryo development [50]. Nevertheless, our study reveals that the parental transmission of the epigenetic imprints generated during clonal propagation could be used as a means of controlling their maintenance or erasure in plant breeding.

In summary, our study has revealed that clonal propagation via somatic embryogenesis using embryonic transcription factors enables the generation of specific epigenomic changes leading to regenerated plants with novel phenotypes. Moreover, because these epigenomic changes are stable albeit exhibit selection/bias during sexual reproduction, stable epimutations could arise and be maintained in populations of plants that alternate between asexual and sexual programs.

## Methods

### Plant growth and clonal propagation

For all experiments, *A. thaliana* Ws-2 plants were grown in long day conditions (16-h light/8-h dark, 22˚C). To facilitate direct plant regeneration from different type of organs we used transgenic lines harboring an chemically-inducible transgenes to overexpress the GRANDE (GRD)/RWPRK motif-containing (RKD4) transcription factor [10] (indRKD4) or the B3 domain transcription factor LEAFY COTYLEDON2 (LEC2) [51] (indLEC2). Seeds from transgenic indRKD4 and indLEC2 lines were germinated on Murashige and Skoog (MS) medium containing 30 μM dexamethasone or 50 μM estradiol, respectively, and incubated for 14 days. Plants were transferred to MS media without dexamethasone or estradiol for 7 days to allow the formation of somatic embryos, which were then isolated and transferred to fresh MS plates using tungsten needles. After 2–3 weeks, the somatic embryos from leaves (LO) or roots (RO) were fully regenerated into new plants. We generated 10 independent regenerants from each organ (G0 generation), which were grown in soil to produce seeds. We grew 24 plants from each line (G1 generation) and selected 10 individuals at random for each regenerated line to produce seed. We continued the sexual propagation of each regenerant population for two additional generations (G2 and G3) following the same scheme.

### Plasmid constructs and transgenic lines

To generate estradiol-inducible RKD4 lines (indRKD4) we cloned using Gateway recombination the coding region of GRD/RKD4 (AT5G53040) in pMDC7 [52] modified by the insertion after PmeI digestion of a pOLE1:OLE1-RFP fragment from pFAST-R [53]. The estradiol-inducible LEC2 lines (indLEC2) were generated by amplifying the LEC2 coding sequence by PCR and cloning in pBI-ΔGR vector. The LEC2-GR fragment was then cloned in pCGN18 [54] to create p35S::LEC2-GR. Plants were transformed by floral dipping, transgenic plants were selected in Kanamycin (indLEC2) or RFP seed selection (indRKD4), and propagated for three generations to isolate homozygous lines.

### Phenotypic analysis

For biomass analysis, plants were grown under long days for 4 weeks (16-h light/8-h dark, 22˚C). Dry weight was recorded using aerial parts excised and dried at 80˚C for 48 hours. For image-based phenotyping, images were acquired once per day in top view using two cameras per tray. The cameras were equipped with OmniVision OV5647 sensors with a resolution of 5 megapixels. Each camera was controlled by a Raspberry Pi computer (Revision 1.2, Raspberry Pi Foundation, UK). Stitched whole-tray images were separated into individual pots using a predefined mask for each position. Segmentation of plant tissue against background was performed by removing the background pixels and applying GrabCut-based automatic post processing. The leaf area of each plant was calculated based on the segmented plant images.

### SA measurement

The total levels of SA and SA glucoside (SAG) were measured from each sample using the *Acinetobacter* sp. ADPWH-*lux* biosensor [55]. In brief, 175 mg of leaves material were harvested from four-week-old plants and homogenized in 250 μl 0.1 M sodium acetate buffer pH 5.5. Samples were then centrifuged for 15 min at 16,000 g. For total SA measurement, 200 μl of supernatant was transferred to a new tube and incubated with 20 μl of 0.5 U/μl β-glucosidase (Sigma-Aldrich) for 90 min at 37˚C. Following β-glucosidase treatment, 60 μl of LB, 30 μl of plant extract, and 50 μl of *Acinetobacter* sp. ADPWH-*lux* (OD = 0.4) were incubated together

at 37˚C for 150 min. Luminescence was read with a Tecan luminometer. SA levels were calculated based on the SA standard curve, which was constructed using known amounts of SA stock (from 0 to 2500 ng/μl) diluted in *NahG* plant extract [56].

## Genome assembly and annotation

Plants were grown in long day conditions (16 h of light) at 23˚C and a relative humidity of 65%. Philips GreenPower TLED modules (Philips Lighting GmbH, Hamburg, Germany) with 110–140 μmol $m^{-2}$ $s^{-1}$ light were used. Plants were grown for 3 weeks. In order to reduce the accumulation of starch all plants were placed in darkness for 30 h before harvesting young leaves. Fresh plant material was immediately flash-frozen after harvesting. HMW DNA extraction was performed following a custom protocol. Approximately thirty grams of leaf tissue were ground in liquid nitrogen and subsequently transferred into a 1 L flask containing 500 mL of ice-cold nuclei isolation buffer. The NIB was composed of 10 mM Tris (pH 8), 0.1 M KCl, 10 mM EDTA (pH 8), 0.5 M sucrose, 4 mM spermidine, and 1 mM spermine-4HCl. Flasks with ground plant material and NIB were kept on ice for fifteen minutes with occasional gentle swirling. The homogenate was filtered through four layers of miracloth (Merck, Germany). Subsequently, NIB solution with 20% Triton X-100 was added. The solution was kept on ice for fifteen minutes with occasional gentle swirling. Afterwards, the solution was distributed into 50 mL Falcon tubes and centrifuged for 15 min at 3,250 rpm at four degrees using a tabletop centrifuge. Nuclei containing pellets were washed by adding a 40 mL NIB solution with 1% Triton X-100 prior to centrifugation for 15 min at 4˚C and 3,250 rpm. Subsequently, the supernatant was discarded before resuspending the nuclei pellet in 30 mL prewarmed (37 C) G2 lysis buffer (Qiagen, Germany). Resuspended pellets were incubated with RNase A (50 μg/ml) for 30 minutes while gently inverting the tube every five minutes. Proteinase K was added (250 μg/ml) and samples were incubated overnight at 50˚C. Subsequently, the lysate was centrifuged at 8,000 rpm at 4˚C for 15 min. The supernatant was poured onto an equilibrated Genomic Tip 100 (Qiagen, Germany). Afterwards the Genomic Tip 100 was washed twice with 7.5 mL QC Buffer (Qiagen, Germany) per wash. Genomic DNA was eluted from the column using 5 mL of prewarmed (50˚C) QF Buffer (Qiagen, Germany). DNA was precipitated by adding 0.7 volumes of isopropanol. HMW DNA was obtained by spooling DNA strings onto a glass hook. The spooled DNA pellet was washed with 80% EtOH prior to resuspending it in 300 μL EB. DNA concentration was measured using Qubit HS DNA kit.

For library construction, high molecular weight DNA was sheared with a Megaruptor (Diagenode) set to a 40 kb fragment size. The DNA library was prepared using the Accel-NGS XL Library Kit (Swift Biosciences, United States). The library was size-selected with a minimum fragment size of 20 kb using a BluePippin (Sage Science, Beverly, United States and a High-Pass v3 cassette. The library was sequenced on the Pacific Biosciences (Menlo Park, United States) Sequel platform (ICS v5.0.0) with the v5.0.1 chemistry, which produced 5.09 Gb of long read sequences with a subread N50 of 15,750 bp.

The genome was assembled into 120 contigs of total length of 123.5 Mb with an N50 of 6.57 Mb using a custom Snakemake pipeline, "auto-asm" (https://github.com/weigelworld/auto-asm). Briefly, long reads were assembled into contigs using canu [57] and scaffolded without polishing steps into chromosomes against TAIR10 using reveal [58]. Whole Genome Annotation was lifted over using the CAT [59] pipeline with TAIR10 as the reference genome and annotation.

## Transcriptome analysis

For mRNA-seq analysis, total RNA (10 μg) for each sample was used to purify polyA+ mRNA that was used for synthesis and amplification of cDNA. The RNA-seq libraries were prepared

using TruSeq RNA Sample Preparation Kit from Illumina (San Diego, CA). RNA libraries were sequenced on an Illumina HiSeq2000 instrument (100 bp single-end). Read quality was assessed using FastQC and trimming of low-quality bases at the 3' end of reads and adapter removal was done using Trimmomatic. Reads were mapped to the TAIR10 reference genome using Tophat (parameters -i 20 -I 30,000) with on average 85.4% unique mappings (min = 82.2, max = 88.8). The read counts for these libraries are given in Dataset S1. We used the R package DESEQ2 (version 1.10.1) [8]. RNA-seq libraries from samples Control-LEC2 root 2 and Control-RKD4 root 3 were discarded for further analysis due to bad quality and low replication fidelity. Genes were classified as significantly differentially expressed at either an FDR<0.01 or <0.05 and |log2 fold-change| >1.5. We used AgriGO v2.0 [60]. The background used for identifying enrichment was the suggested background provided by AgriGo, which contains all GO annotated genes in Arabidopsis. Enrichment was performed using a hypergeometric test, p-values were corrected by multiple correction (Benjamini Hochberg FDR; α = 0.05).

## Identification of polymorphisms in RO-LEC2 lines

For library construction, high molecular weight DNA from individual plants was isolated using DNeasy Plant mini kit (Qiagen). The DNA library was prepared using a NxSeq Amp-FREE Low DNA Library kit (Lucigen) and sequenced on a Illumina HiSeq2000 with a 100bp paired-end mode. Sequencing reads were pre-processed using fastp with default settings and the overrepresentation analysis parameter to remove reads with low quality score, irregular GC content, short length, and sequencing adapters present [61]. Trimmed reads were then quality checked with outputs from fastp to ensure trimming had been successful. Trimmed reads were mapped to the TAIR10 reference genome using bowtie2 [62]. Variant calling was performed on aligned and sorted BAM files using samtools mpileup and piped to bcftools to produce VCF files [63]. VCF files were converted to SHOREmap format and analysed using SHOREmap backcross and then annotated to output tables and plots of backcross SNPs [64].

## Bisulfite sequencing

Bisulfite libraries were prepared from 100 ng genomic DNA using the TruSeq Nano kit (Illumina, San Diego, CA, USA) according to the manufacturer's instructions, with the following modification. After adapter ligation and cleanup, samples were bisulfite-treated using the Epitect Plus kit (Qiagen, Hilden, Germany). Thermal cycle incubation was repeated once before clean-up. DNA fragments were amplified using the Kapa HiFi Hotstart Uracil+ PCR mix in 14 cycles of PCR (Kapa Biosystems, Wilmington, MA, USA). Bisulfite libraries were sequenced on an Illumina HiSeq3000 instrument (150 bp paired-end reads).

## Computational analysis of paired end BS-seq

Paired-end quality was assessed using FASTQC [65]. Trimmomatic [66] was used for quality trimming. Parameters for read quality filtering were set as follows: Minimum length of 40 bp; sliding window trimming of 4 bp with required Phred quality score of 20. Trimmed reads were mapped to the *Arabidopsis thaliana* TAIR10 genome assembly using bwa-meth [67] with default parameters. Mapped reads were deduplicated using picardtools [68], and numbers of methylated/unmethylated reads per position were retrieved using MethylExtract [69]. We calculate differentially methylated regions (DMRs) using MethylScore [70].

### Data visualization

For visualizing BS-seq genomic data we used Integrative Genomic Viewer (IGV) [71]. For figures, we used R version 3.5.1 (www.r-project.org) with packages ggplot2 [72], eulerr [73] and pheatmap [74].

## Supporting information

**S1 Fig. Arabidopsis Ws-2 plants from clonal RO-LEC2 and LO-LEC2 G3 progeny grown under different growth conditions.**
(EPS)

**S2 Fig. DEGs in LO-RKD4 and RO-RKD4 regenerants compared to Ws-2/indRKD4 control plants.** Data from three randomly selected regenerants for each tissue.
(EPS)

**S3 Fig. DEGs in LO-LEC2 and RO-LEC2 regenerants compared to Ws-2/indLECs control plants.** Data from three randomly selected regenerants for each tissue.
(EPS)

**S4 Fig. DNA methylation variation in LEC2 regenerants.** (A). CG methylation levels at 226 CG-DMRs identified in an all-against-all comparison. (B). CHG methylation levels at 120 CHG-DMRs identified in an all-against-all comparison. (C). CHH methylation levels at 386 CHH-DMRs identified in an all-against-all comparison.
(EPS)

**S5 Fig. DNA methylation variation in LEC2 regenerants affects all contexts and appears mainly in proximity to genes.** (A). PCAs of CHG DNA methylation levels at 120 CHG-DMRs identified in all-against-all comparison (left two panels) and PCAs of CHH methylation levels at 386 CHH-DMRs identified in all-against-all comparison (right two panels). Percentages in brackets indicate the proportion of variance explained by each principal component (PC). (B) Distribution of different genetic features in DMRs identified in all-against-all comparison for all three contexts, compared to regions identified as methylated in each sample and to the distribution of features across the whole genome. Intergenic is defined as all sequences more than 2 kb away from a gene.
(EPS)

**S6 Fig. Distribution of SNPs associated with early senescence of RO-LEC2 plants.** Allelic frequencies across chromosomes of foreground markers derived from comparing phenotypically positive (foreground) against phenotypically negative (background) plants.
(EPS)

**S7 Fig. Hypermethylated RO-LEC2 DMRs revert to normal methylation levels after backcrossing.** Heatmaps showing gains and losses of CHG (A) and CHH (B) DNA methylation levels for RO-LEC2 leaf DMRs in $F_2$ progeny from reciprocal backcrosses between control (Ws-2/indLEC2) and RO-LEC2 plants. Each column represents plants from independent $F_2$ progenies.
(EPS)

**S8 Fig. Epigenome browser view of AtSIN4 genomic region showing DNA methylation, ATAC-seq, H3K27me3 and Hi-C data from roots and shoots.**
(EPS)

**S9 Fig. Expression of *PRE1/BNQ1* is associated with the hypomethylation of a flanking AtMUNX1 transposon.**
(EPS)

**S1 Table. Differentially Expressed Genes in Ws-2/LEC2 regenerated lines.**
(XLSX)

**S2 Table. Differentially Expressed Genes in Ws-2/RKD4 regenerated lines.**
(XLSX)

**S3 Table. List of Differentially Methylated Regions at Differentially Expressed Genes in Ws-2/LEC2 regenerated lines.**
(XLSX)

## Acknowledgments

We thank Gary Grant for help with plant husbandry, and Liliana M. Costa for discussions and comments on the manuscript.

## Author Contributions

**Conceptualization:** Anjar Tri Wibowo, Detlef Weigel, Jose Gutierrez-Marcos.

**Data curation:** Anjar Tri Wibowo, Javier Antunez-Sanchez, Alexander Dawson, Jonathan Price, Cathal Meehan, Travis Wrightsman, Maximillian Collenberg, Ilja Bezrukov, Claude Becker.

**Formal analysis:** Anjar Tri Wibowo, Javier Antunez-Sanchez, Alexander Dawson, Jonathan Price, Cathal Meehan, Travis Wrightsman, Maximillian Collenberg, Ilja Bezrukov, Claude Becker, Moussa Benhamed.

**Funding acquisition:** Detlef Weigel, Jose Gutierrez-Marcos.

**Investigation:** Anjar Tri Wibowo, Javier Antunez-Sanchez, Claude Becker.

**Project administration:** Detlef Weigel, Jose Gutierrez-Marcos.

**Writing – original draft:** Anjar Tri Wibowo, Javier Antunez-Sanchez, Detlef Weigel, Jose Gutierrez-Marcos.

**Writing – review & editing:** Anjar Tri Wibowo, Javier Antunez-Sanchez, Detlef Weigel, Jose Gutierrez-Marcos.

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
