## [Decision Letter · Decision Letter 0]

11 Jun 2022

Dear Dr Gutierrez-Marcos,

Thank you very much for submitting your Research Article entitled 'Predictable and stable epimutations induced during clonal plant propagation with embryonic transcription factors' to PLOS Genetics. I apologize for the lengthy review process as there was some difficulty in obtaining the reviews in a timely fashion.

The manuscript was fully evaluated at the editorial level and by independent peer reviewers. The reviewers appreciated the attention to an important problem and all expressed enthusiasm for aspects of this study, but several reviews raised some concerns about the current manuscript. Based on the reviews, we will not be able to accept this version of the manuscript, but we would be willing to review a revised version. We cannot, of course, promise publication at that time.

Two of the reviewers made helpful suggestions or comments that could be incorporated to improve the manuscript while the third reviewer expressed enthusiasm without requesting any revisions. Some of the suggestions of reviewer 2 might be beyond the scope of the current study and this is acknowledged within the review.  However, it might be good to use these comments and revising some of the presentation of the data. Reviewer 1 made some good points about the descriptions of phenotypes and the comparisons of the different TFs that were utilized. Additional clarity in the descriptions of the experiments will help to clearly communicate the findings and interpretations.

If you decide to revise the manuscript for further consideration at PLOS Genetics, please aim to resubmit within the next 60 days, unless it will take extra time to address the concerns of the reviewers, in which case we would appreciate an expected resubmission date by email to plosgenetics@plos.org.

[LINK]

We are sorry that we cannot be more positive about your manuscript at this stage. Please do not hesitate to contact us if you have any concerns or questions.

Yours sincerely,

Nathan M. Springer

Associate Editor

PLOS Genetics

Claudia Köhler

Section Editor: Plant Genetics

PLOS Genetics

Reviewer's Responses to Questions

**Comments to the Authors:**

Reviewer #1: In previous work, the authors made the important and really interesting observation that tissue/organ-specific DNA methylation states can be stably inherited in plants regenerated from asexual somatic embryogenesis, and that these methylation states even persist in tissues that usually have a different methylation state (Wibowo et al 2018).

In this work the authors were motivated to explore this phenomenon further, “in particular the importance of the genetic background and/or the transcription factor employed”. In this work, the Ws background was used and inducible LEC2 line is the focus.

Findings:

- Plants regenerated from roots had distinctly less biomass and early senescence (LEC2 only?).

- Differential gene expression in root origin plants was enriched for GO categories for defence, senescence and stress; and these plants had high levels of SA in both LEC2 and RDK4 plants

- Tissue of origin DNA methylation imprints observed (consistent with in prior work), this time mainly root hypomethylation alleles (LEC2)

- Identified candidate epialleles associated with the early senescence phenotype and hypothesis this may be due to loss of CHH at a TE

- It was also interesting to note that in root origin plants, hypermethylated loci were only stably maintained maternally and methylation was lost during paternal transmission

Overall, it was useful to conduct the replication of the tissue of origin epigenetic imprints in another genetic background using a different TF for somatic embryogenesis. The early senescence phenotype is also quite compelling, although whether it is caused by an epigenetic change was not fully established (could it be a structural change eg a TE that jumped?), even though some candidates were proposed. Future work is required to resolve the cause of this phenotype.

Major:

One of the themes of this paper is that the phenotypic and molecular imprints in regenerated plants were independent of the TF used for somatic embryogenesis. The paper was mostly focused on the LEC2 line, which I think is not a problem; but, in particular for the early senescence and reduced biomass of RO plants; it should be clarified if this was also observed in RKD4 plants? The RO-RKD4 plant in Fig 1B looks like the control (?) and the data in Fig1C-D is only for LEC2? Can the data for RKD4 be presented too, or is this phenotype specific for LEC2?

Minor:

- it would be good to title the Results section “Results”!

- please clarify if G0 is the regenerated generation (page 4) (I assume so but it is not stated)

- page 4 it says 4/5 RO plants Fig1C-D - but there are 10 RO plants in Fig1C???

- The conclusion on Page 4 is that tissue of origin is the major determinant of phenotypic variation. I agree the phenotypes are impressive. But this conclusion seems too broad as the data are only for select phenotypes, I’d suggest revising to something more modest. (“a major contributor… to the phenotypes we measured”). Also, as above in the Major comment, Fig1C-D only includes data for LEC2, please include the data for RKD4 to support this statement or how do we know there isn’t a strong tissue x TF effect?

- page 5 (or in methods) please state how many biological replicates were used for RNA-seq. Looking at the supp tables, does one group (WT-root) have only 2 replicates? (on a side note, why still using Tophat!... 14 years old now)

- In the discussion about DEGs on page 5 it would be helpful to include the number(s) of DEGs in text when highlighting that some contrasts have more than others.

- Fig 2C, D and E fonts are too small to read.

- Is Fig2F RNA-seq (or eg qPCR)? Are the differences statistically significant, please add stats to this figure?

- Please explain the axis categories in Fig3-S2 more clearly. “DMRs” = all DMR loci? But what does control, LO-G2 etc refer to; are these the DMRs in each genotype or is this all regions with methylation data coverage?

- Page 13 “...although they were faithfully inherited maternally (Fig4A)” should be Fig4B

- Is the Ws assembly available somewhere?

Reviewer #2: This interesting manuscript describes the epigenetic impact of clonal propagation via somatic embryogenesis of Arabidopsis (Ws-2) using the embryonic transcription factors RDK4 and LEC2. The authors show that plants regenerated from roots (RO) were noticeably senesced earlier than control plants or plants regenerated from shoots. This phenotype appeared more pronounced for LEC2 than RDK4, and remained stable over 3 generations. Further characterisation revealed that progeny from root-regenerated LEC2-plants (LEC2-RO) plants contained higher concentrations of salicylic acid (SA), showed gene expression patterns characteristic for senescence and immune responses to pathogens. These leaf phenotypes were associated with genome-wide DNA hypomethylation, which resembled the methylation profile of roots in control plants. To further confirm that the somaclonal variation is under control by stably inherited epigenetic DMRs, the authors performed reciprocal crosses between Ws-2 and RO-LEC2 plants and found that the segregation pattern of early senescence resembled that of a single recessive locus. Subsequent methylome analysis of 20 F2 segregants revealed 36 DMRs co-segregating with the early senescence phenotype. The authors compete their impressive study by describing the epigenetic and transcriptomic status of two genomic regions where methylation status co-segregated with the phenotype for both directions of the crosses.

Overall, I was deeply impressed by this study. While the discovery of stably inherited somaclonal epigenetic variation is not new, this paper provides a thorough mechanistic characterisation of this epigenetic variation over multiple generations of inbreeding and outcrossing. As is common for excellent research, the study raises many questions and suggestions for follow-up research (I have listed a few below) However, these merely reflect my enthusiasm for this study and I don’t feel these questions should necessarily be addressed with further experimentation in order to merit publication of this excellent study in PLoS Genetics.

Questions/comments:

1. The study stops short of demonstrating a causal relationship between the two DMRs and the senescence phenotype. Is there are evidence in the Col-0 that genetic mutations in the genes flanking the DMRs cause senescence / disease resistance phenotypes?

2. The authors state the leaf phenotypes of LEC2-RO plants are associated with genome-wide DNA hypomethylation, which resembles the methylation profile of these regions in roots from control plants. However, roots do not necessarily accumulate increased levels of SA – if anything they accumulate much lower levels of SA than leaves. (see e.g. Bagautdinova et al. Int J Mol Sci. 2022 Feb; 23(4): 2228). Can the authors provide an explanation why the root-derived epigenetic state in leaves results in SA hyperaccumulation, while the same epigenetic state in roots doesn’t?

3. Based on the transcriptome analysis of LEC2-RO plants, the authors assume that the sae epialleles control both senescence and SA hyperaccumulation. Can this be confirmed by the phenothpes of the F2 segregants? In other words, does the senescence phenotype co-segregate with increased SA levels and/or defence-related gene expression?

4. Related to the above points, total SA levels (free SA and SA glucoside) were quantified by the ADPWH-lux biosensor, which doesn’t differentiate between free SA and glycosylates forms of the SA. This is somewhat unfortunate as a disturbed balance between free and conjugated SA might explain the senescence phenotypes. It would e great it the authors could address this point in the Discussion and/or provide extra data where possible.

5. Top of 14: “4F [will change to G])”. Is this a left-over comments from the authors’s own revisions?

6. Figure 4E the different DNA methylation categories would benefit from a but more explanation than the somewhat cryptic information provided in the legend.

7. Some of the sequencing tracks of the supplemental figures painfully reminded me to upgrade my reading glasses as they are labelled by very small letter fonts.

Reviewer #3: This is a well-written and comprehensive manuscript that describes heritable variation associated with the tissue from which clones are produced. The thorough analysis of sequence and epigenome data supports the conclusions that heritable variation is based - at least in significant part - on pre-existing patterns in the originating tissues. A reasonable hypothesis is advanced on the basis of the phenotype(s) observed over generations. The manuscript is very well done.

**Have all data underlying the figures and results presented in the manuscript been provided?**

Reviewer #1: Yes

Reviewer #2: Yes

Reviewer #3: Yes

PLOS authors have the option to publish the peer review history of their article (what does this mean?). If published, this will include your full peer review and any attached files.

Reviewer #1: No

Reviewer #2: **Yes: **Jurriaan Ton

Reviewer #3: No

---

## [Decision Letter · Decision Letter 1]

15 Oct 2022

Dear Dr Gutierrez-Marcos,

We are pleased to inform you that your manuscript entitled "Predictable and stable epimutations induced during clonal plant propagation with embryonic transcription factors" has been editorially accepted for publication in PLOS Genetics. Congratulations!  Thank you for the careful and clear responses to reviews.

Yours sincerely,

Nathan M. Springer

Academic Editor

PLOS Genetics

Claudia Köhler

Section Editor

PLOS Genetics

Comments from the reviewers (if applicable):

Reviewer's Responses to Questions

**Comments to the Authors:**

Reviewer #1: All my queries have been clearly addressed, thank you.

Reviewer #2: The authors have addressed all my questions and comments.

**Have all data underlying the figures and results presented in the manuscript been provided?**

Reviewer #1: Yes

Reviewer #2: Yes

PLOS authors have the option to publish the peer review history of their article (what does this mean?). If published, this will include your full peer review and any attached files.

Reviewer #1: No

Reviewer #2: **Yes: **Jurriaan Ton

**Data Deposition**

http://datadryad.org/submit?journalID=pgenetics&manu=PGENETICS-D-22-00513R1

**Press Queries**

---

## [Editor Report · Acceptance letter]

9 Nov 2022

PGENETICS-D-22-00513R1 

Predictable and stable epimutations induced during clonal plant propagation with embryonic transcription factors 

Dear Dr Gutierrez-Marcos, 

We are pleased to inform you that your manuscript entitled "Predictable and stable epimutations induced during clonal plant propagation with embryonic transcription factors" has been formally accepted for publication in PLOS Genetics! Your manuscript is now with our production department and you will be notified of the publication date in due course.

With kind regards,

Bernadett Koltai

PLOS Genetics

On behalf of:
